# Phase Angle Trajectory Among Critical Care Patients: Longitudinal Decline Predicts Mortality Independent of Clinical Severity Scores

**DOI:** 10.3390/healthcare13121463

**Published:** 2025-06-18

**Authors:** Pantelis Papanastasiou, Stavroula Chaloulakou, Dimitrios Karayiannis, Avra Almperti, Georgios Poupouzas, Charikleia S. Vrettou, Vasileios Issaris, Edison Jahaj, Alice G. Vassiliou, Ioanna Dimopoulou

**Affiliations:** 1Department of Clinical Nutrition, Evangelismos General Hospital, 10676 Athens, Greece; ppapanasta@gmail.com (P.P.); st.chaloulakou@gmail.com (S.C.); aalmperti@yahoo.gr (A.A.); 2Pulmonary Department, Evangelismos General Hospital, 10676 Athens, Greece; poupouzas.gewr.kw@gmail.com (G.P.); vasilisiss@gmail.com (V.I.); 31st Department of Critical Care Medicine, School of Medicine, National and Kapodistrian University of Athens, Evangelismos Hospital, 10676 Athens, Greece; vrettou@hotmail.com (C.S.V.); edison.jahaj@gmail.com (E.J.); alvass75@gmail.com (A.G.V.); idimo@otenet.gr (I.D.)

**Keywords:** bio-impedance analysis, ICU, long-term mortality, phase angle, APACHE II score, SOFA score

## Abstract

Background/Objectives: The phase angle (PhA) is an emerging biomarker reflecting the cellular integrity and nutritional status. This study aimed to explore potential associations between the PhA, clinical severity scores, and 60-day survival outcomes following an admission to the Intensive Care Unit (ICU). Methods: This prospective, single-center study included 43 critically ill patients admitted to the ICU at Evangelismos General Hospital between May and November 2024. Patients were stratified by their PhA (≤5.4° vs. >5.4°). The PhA was measured at admission and subsequently on days 5–7, 10–11, 13–14, and until discharge. Severity scores (SOFA and APACHE II) were recorded. Between-group differences were assessed using independent samples *t*-tests and Mann–Whitney U tests, as appropriate. Survival was analyzed using Kaplan–Meier curves and Cox proportional hazards models. Results: The mean age was 54.6 ± 17 years; 63.6% were male. At ICU admission, patients with a PhA > 5.4° were significantly younger (*p* < 0.001) and had a higher fat-free mass (*p* < 0.001), greater calf circumference (*p* < 0.001), higher extracellular water (*p* < 0.001), larger mid-upper arm circumference (*p* = 0.009), and higher resting energy expenditure per kilogram (27.4 vs. 23.1 kcal/kg, *p* = 0.002). The PhA declined significantly during the ICU stay (*p* < 0.001). The Kaplan–Meier analysis showed a significantly shorter survival in patients with a PhA ≤ 5.4° (HR: 6.32, *p* = 0.019), which remained significant after adjusting for sepsis (*p* = 0.017). In a multivariable Cox regression, both PhA and APACHE II scores independently predicted mortality. Conclusions: While limited by a small sample size and single-center design, these findings support the further exploration of the PhA as a monitoring tool in critical care.

## 1. Introduction

The phase angle (PhA) is a biomarker that reflects the cellular integrity, hydration status, and overall nutritional health [1]. PhA is derived from a bioelectrical impedance analysis, and it describes the angular difference (phase difference) between the voltage and current sinusoidal waveforms [2]. In the human body the current reaches its maximum and minimum peak after the voltage (positive values), due to the presence of cell membranes and tissue interfaces [2]. In general, the PhA is positively associated with muscle strength and fluid distribution, therefore a higher PhA indicates a better cell membrane integrity and function, and it is higher in athletes, while it declines with age [2,3]. In critical illness, as well as other conditions, like inflammation, malnutrition, and inactivity, the PhA decreases and leads to a reduced quality of life and poor outcomes [2,3]. According to ESPEN guidelines for ICU patients, any ICU patient with a length of stay longer than 48 h is considered at risk for malnutrition [4]. Given these associations, there is growing interest in the potential of the PhA as a prognostic tool in acute clinical settings, such as the Intensive Care Unit (ICU) [5,6,7].

In the ICU, clinical severity is commonly assessed using standardized scoring systems, such as the Acute Physiology and Chronic Health Evaluation II (APACHE II) and the Sequential Organ Failure Assessment (SOFA) [8]. The APACHE II score incorporates physiological variables, laboratory values, and information about chronic health conditions to estimate the risk of mortality [9]. The SOFA score evaluates the extent of the organ dysfunction across multiple systems, including respiratory, cardiovascular, hepatic, coagulation, renal, and neurological functions [10]. These tools are widely used and considered in treatment plan decisions, and they also seem to predict patient outcomes [11,12]. However, they may not fully account for nutritional status or cellular health, which are emerging as important prognostic factors in critical illness, and they may be less accurate for a long-term diagnosis [13]. The PhA has gained attention as a potential complementary biomarker, as its clinical relevance lies in its ability to reflect nutritional and cellular health in real-time, using a non-invasive and repeatable method. Thus, it offers potential value for longitudinal monitoring in critically ill patients [13].

However, both scoring systems have limitations. They provide a snapshot of the moment, may underrepresent nutritional and cellular factors, and have a limited capacity to monitor ongoing physiological changes or predict long-term outcomes [13]. Therefore, there is a need for complementary measures that capture dynamic, long-term changes in the health status throughout the ICU stay.

Despite a high interest, a gap remains in the literature regarding how the PhA can be practically integrated with traditional severity scores.Given the growing body of evidence and the need for more comprehensive prognostic tools in critical care, this study aimed to investigate the association between the PhA measured on ICU admission and clinical severity scores, body composition parameters, metabolic alterations, and 60-day survival outcomes in a cohort of critically ill patients. By exploring these associations, we sought to determine whether the PhA could enhance current prognostic models and serve as a practical tool for risk stratification and clinical monitoring in the ICU.

## 2. Materials and Methods

### 2.1. Criteria

We conducted a prospective observational study including adult patients admitted to the 1st Intensive Care Unit of Evangelismos Athens General Hospital, a tertiary care center, between May and November 2024. Eligible participants were adults aged 18 years or older who expected to remain in the ICU for more than 48 h. Exclusion criteria included pregnancy, presence of pacemakers or other electronic implants, edema that could affect bioelectrical impedance analysis accuracy, terminal illness with expected survival less than 48 h, limb amputation, pre-existing neuromuscular diseases affecting body composition, and corticosteroid use at admission. Patients readmitted to the ICU during the same hospital stay were also excluded. PICO (patient/population, intervention, comparison, and outcome) criteria are presented in Table 1.

This study was approved by the Institutional Review Board of Evangelismos Hospital of Athens (Approval Number: 467/11-1-2023, Study of ICU patients with brain injury: the role of glucocorticoid receptors and newer brain injury biomarkers on their course, prognosis, and outcomes) and was conducted in accordance with the ethical principles of the Declaration of Helsinki. Informed written consent was obtained from all patients’ next-of-kin.

### 2.2. Measures

All measurements were conducted by a trained clinical dietitian on admission, and thereafter, at days 5–7 (second time-point), 10–11th day (third time-point), 13–14th day (fourth time-point), and until discharge from the hospital. Upon ICU admission, demographic data, anthropometric measurements, and metabolic parameters were recorded. To determine the optimal prognostic cut-off value for PhA at ICU admission, a maximally selected rank statistics analysis was performed. This method identifies the cut-point that maximizes the difference in survival between patient groups based on standardized log-rank statistics. The analysis established a cut-off point of 5.4°, which provided the greatest separation in survival outcomes (Figure 1). For survival and clinical outcome analyses, patients were subsequently stratified into two groups: low PhA (≤5.4°) and high PhA (>5.4°). For further multivariable survival analysis, to avoid subjectivity, PhA was inserted as a continuous variable. Clinical severity was assessed using the SOFA and APACHE II scores. Body composition markers included body fat, fat-free mass, mid-upper arm circumference (MUAC), calf circumference, and extracellular water volume. Resting energy expenditure (REE) was measured using indirect calorimetry and normalized per kilogram of body weight (REE/kg).

BIA (bioelectrical impedance analysis) was conducted using a multifrequency analyzer according to standardized protocols, with patients in a supine position. PhA was calculated from resistance and reactance values at 50 kHz and recorded in degrees. BIA is based on the electrical principle that the body is a circuit with a given R and Xc. R reflects the opposition of current flow through intracellular and extracellular solutions, and Xc reflects the capacitance of the cells to store energy. PhA is an indicator of overall cellular health, membrane integrity, and body cell mass, calculated from BIA. It is derived as the arctangent of the ratio between reactance (Xc) and resistance (R), reflecting the phase shift between voltage and current as an electrical signal pass through body tissues. This value represents how efficiently cells store electrical charge, with higher PhA values indicating better cellular function and membrane integrity. The reliability and validity of BIA has been justified elsewhere [14].

PhA values can vary naturally based on individual factors, such as age, sex, and body composition. Moreover, in critically ill patients, changes in hydration status, inflammation, and tissue integrity can influence the body’s impedance characteristics, altering both resistance and reactance. As a result, shifts in PhA during critical illness may reflect dynamic changes in cellular health and nutritional status, making it a valuable, non-invasive marker for monitoring clinical progress and prognosis [15].

### 2.3. Primary and Secondary Outcomes

The primary outcomes of this study were PhA, APACHE II, and SOFA. These outcomes were used to evaluate the prognostic and clinical relevance of PhA in critically ill patients. Secondary outcomes included 60-day all-cause mortality; body composition parameters (fat-free mass, body fat, mid-upper arm circumference, calf circumference, and extracellular water); metabolic indicators, such as resting energy expenditure per kilogram of body weight (REE/kg); and the longitudinal trajectory of PhA during the ICU stay.

### 2.4. Statistical Analysis

Statistical analyses included independent *t*-tests, Mann–Whitney U tests, and χ^2^ tests for group comparisons. Longitudinal changes in PhA were analyzed using Durbin Conover pairwise comparisons. Survival outcomes were assessed using Kaplan–Meier analysis, and predictors of mortality were identified through multivariate Cox proportional hazards regression, adjusting for sex, sepsis, APACHE, SOFA scores, medication, and body fat (kg).

A priori power analysis was performed to determine the minimum required sample size for detecting a statistically significant difference in PhA values from a predefined reference value in critically ill ICU patients. Assuming a moderate effect size (Cohen’s d = 0.500), a two-tailed significance level (α) of 0.05, and a desired statistical power of 80%, the estimated required sample size was 34 participants. The sample size calculation was conducted using R software version 4.4.0. This sample size estimation ensured adequate power for comparing PhA values against the prognostic threshold identified in previous studies.

## 3. Results

### 3.1. Patients

A total of 43 critically ill patients were included in this study, with a mean age of 54.6 ± 17 years, and 63.6% were male. The most common etiologies included polytrauma, septic shock, pneumonia, subarachnoid hemorrhage (aSAH), and postoperative monitoring following major surgeries such as a Whipple procedure, hepatectomy, or colectomy. Baseline comparisons between groups are summarized in Table 2.

### 3.2. Outcomes

Upon the ICU admission, patients with a PhA > 5.4° were significantly younger (*p* < 0.001) and had a lower body fat (*p* = 0.008), higher fat-free mass (*p* < 0.001), greater calf circumference (*p* < 0.001), and higher MUMC (*p* = 0.009) compared to those with a PhA ≤ 5.4°. They also demonstrated a higher resting energy expenditure per kilogram of body weight (REE/kg) (27.4 vs. 23.1, *p* = 0.002).

Longitudinal monitoring revealed a significant decline in the PhA over time, with median values decreasing from 5.45 at baseline to 5.15 at the final measurement point (13–14th day) (*p* < 0.001), Figure 2.

The Kaplan–Meier analysis demonstrated a significantly lower 60-day survival among patients with a PhA ≤ 5.4° (Hazard Ratio [HR]: 6.32, *p* = 0.019), as presented in Figure 3. Patients with a higher PhA exhibited significantly better long-term survival outcomes, with survival rates of 92.31% at both 12 and 36 months and 80.77% at 60 months (survival time rmean: 61.6 months). In contrast, patients with a low PhA showed a rapid decline in survival over time: 91.67% at 12 months, decreasing sharply to 55.56% at 36 months, and reaching 0% survival by 60 months(survival time rmean: 31.9 months).

This association remained statistically significant in the multivariate model even after adjusting for sepsis (HR = 10.11, 95%, *p* = 0.017), Figure 4. Kaplan–Meier survival curves showed that patients with a high PhA (>5.4) and no sepsis had the best survival outcomes, with no deaths observed during the follow-up period (solid blue line). In contrast, those with a low PhA (≤5.4) and no sepsis showed a substantially worse survival (dashed blue line), with a marked decline after the first week and no survivors beyond 36 days (log-rank *p* = 0.027). Notably, patients with a low PhA and sepsis (green dashed line) had similarly poor outcomes, with the survival dropping to 0% by day 42. Patients with a high PhA and sepsis (red dotted line) showed an intermediate survival, with a moderate decline but no deaths past day 36.

Furthermore, a multivariable cox regression analysis was conducted to investigate factors associated with survival, using the length of stay as the time variable. The PhA was entered into the model as a continuous variable, and it was found to be significantly associated with survival. A higher PhA was protective, with a HR of 0.10 (95% CI: 0.01–0.76, *p* = 0.049). In addition, the APACHE score was a significant predictor, with a HR of 1.59 (95% CI: 1.01–2.50, *p* = 0.04), indicating an increased risk with a higher score (Table 3).

## 4. Discussion

The findings of this study highlight the prognostic significance of the PhA in critically ill patients. Our study showed that a higher PhA on ICU admission is associated with better body composition parameters, including a higher fat-free mass, muscle circumference, and higher resting energy expenditure per kg. Conversely, patients with a lower PhA exhibited a significantly higher mortality risk, independent of traditionally used clinical severity scores such as the APACHE II and SOFA.

The best survival outcomes were observed in patients with a PhA > 5.4 and no sepsis, while those with both a low PhA and sepsis had the poorest prognosis. Among the variables analyzed in the multivariable cox regression analysis, the PhA demonstrated a strong protective effect against mortality. The inverse relationship suggests that patients with higher PhAs are more likely to survive, possibly due to their better overall cellular health and nutritional reserves. Finally, the APACHE II score retained its predictive validity in the multivariate survival analysis, reinforcing its role in clinical decision-making. While other variables, such as sex, SOFA scores, body fat, and medication, did not reach statistical significance in this model, body fat showed potential clinical relevance. Notably, body fat exhibited a trend toward a protective association which could support the concept of the “obesity paradox” seen in some critical care studies. However, it is important to note that the SOFA score was not significantly associated with mortality, which may reflect its limited sensitivity in predicting long-term outcomes or capturing nutritional and cellular deterioration.

Our findings align with those of Rosenfeld et al. [7], who demonstrated that lower PhA values were significantly associated with an increased mortality in critically ill older adults, both during their ICU stay and at 28 and 60 days post-admission. The PhA emerged as a stronger predictor of late mortality than traditional clinical severity scores. On the same page, Garcia-Grimaldo et al. [16] demonstrated that a decline in the PhA over 14 days was significantly associated with an increased 90-day mortality in critically ill COVID-19 patients. Similarly to their results, we found that longitudinal PhA trends offered more prognostic insight than single-time-point measurements. Similarly, another study by Stapel et al. [17] showed an independent association between the PhA and 90-day mortality in 196 ICU patients. Their findings showed that patients with a lower PhA on admission had significantly higher mortality rates, even after adjusting for the APACHE II scores, age, sex, and BMI. Notably, their identified optimal cut-off of <4.8° closely aligns with the threshold used in our cohort. A similar cut-off point has also been set by Kereski da Silva et al. [5], who reported a PhA cut-off of 5.1° as predictive of worse outcomes in critically ill patients. While their study observed stronger correlations in non-septic patients, our data suggests a broader applicability across varied ICU etiologies, including septic and non-septic cases. This consistency in cut-off values across populations supports the potential for the PhA to serve as a standardized prognostic marker in critical care.

A recent systematic review and meta-analysis by Lima et al. [6], encompassing 4872 critically ill patients across 27 studies, also confirmed that lower PhA values were associated with an increased mortality and longer ICU stays. The pooled relative risk of death for patients with a low PhA was 1.82, and survivors consistently showed higher PhA values. Despite some variability in study methodologies and patient characteristics, this meta-analysis reinforced the prognostic value of the PhA as a clinically meaningful, non-invasive tool for mortality predictions in ICU settings.

Importantly, our study adds to the literature by tracking longitudinal changes in the PhA during ICU stays. We observed a progressive decline in the PhA from admission to the fourth measurement point, consistent with the catabolic state, inflammation, and fluid shifts commonly experienced by critically ill patients. The capacity of the PhA to reflect both the baseline nutritional and cellular integrity and dynamic changes during illness progression suggests its value not only as a baseline risk marker but also as a real-time monitoring parameter.

Current ESPEN guidelines for clinical nutrition in the Intensive Care Unit recommend a bioelectrical impedance analysis as a supportive tool for assessing the body composition and hydration status in critically ill patients, acknowledging its limitations in the presence of fluid shifts and inflammation [4]. While the PhA is recognized as a parameter of interest, it is not yet routinely recommended for prognostic assessment. However, our findings, along with those of recent studies, suggest that the PhA holds significant prognostic potential, warranting further investigation and possible future integration into ICU care algorithms. Despite this promise, practical challenges remain. These include standardizing measurement protocols across operators, interpreting values in fluid-overloaded or septic patients, and determining how to incorporate PhA trends into existing care pathways. Additionally, clinicians may require training to interpret PhA data effectively in real-time.

The association between a low or declining PhA and worse outcomes in critically ill patients may be explained by the underlying physiological mechanisms that the PhA reflects. The PhA is derived from bioelectrical impedance and serves as a marker of cellular integrity and the body cell mass. Lower PhA values may indicate disrupted cell membrane functions and a reduction in cell mass, both of which are common in catabolic states, inflammation, and malnutrition. Additionally, inflammation and sepsis can lead to capillary leakage and fluid shifts, increasing extracellular water and thereby lowering the PhA. Thus, a low or declining PhA may reflect the severity of systemic illness, muscle wasting, and nutritional deterioration, which contribute to impaired recovery and increased mortality [6].

This study’s limitations include its relatively small sample size, which may limit the generalizability of the results. Additionally, the selected cut-off for the PhA, while meaningful in this cohort, may not be universally applicable. Despite these constraints, the significant association between a low PhA and mortality highlights the potential utility of this simple, non-invasive measurement. Moreover, the influence of potential confounding variables, such as the hydration status, was not fully controlled. While we accounted for the use of medications, detailed fluid therapy metrics and inflammation markers were not consistently accounted for in our analysis. This limitation reflects a broader issue in critical care research, as the fluid balance and inflammatory status are often difficult to standardize in ICU settings. These factors can impact the PhA independently of cellular health and should be accounted for in future studies.

Despite these limitations, our results reinforce the potential utility of the PhA as a simple, bedside, non-invasive biomarker for risk stratification and clinical monitoring in the ICU. Future research should focus on validating PhA cut-off points across larger, multi-center cohorts and diverse patient populations. Additionally, prospective interventional studies investigating whether tailored nutritional or rehabilitation strategies based on PhA trends can improve clinical outcomes are warranted.

## 5. Conclusions

This study is among the first to highlight the clinical relevance of longitudinal PhA tracking in critically ill and septic patients. PhA assessments may offer a practical non-invasive tool to monitor trajectory, identify early deterioration, and support individualized nutritional and therapeutic interventions in the ICU.

## Figures and Tables

**Figure 1 healthcare-13-01463-f001:**
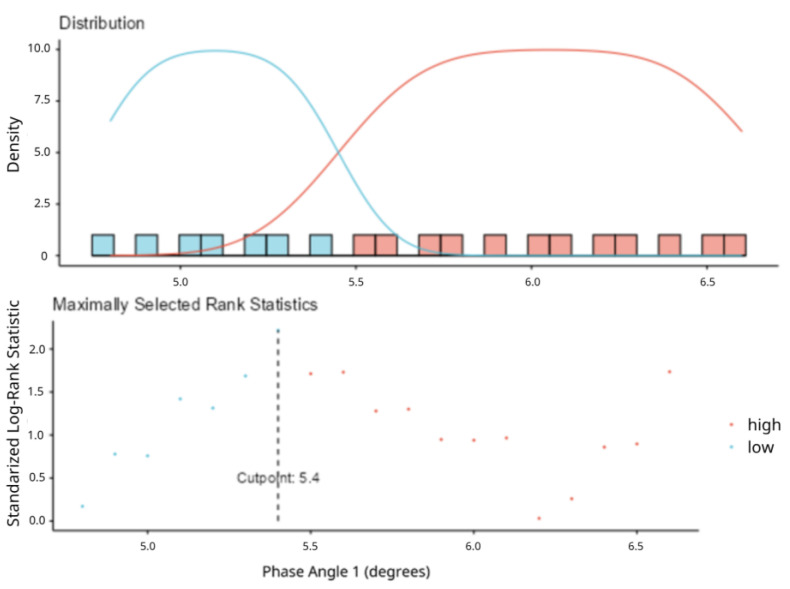
The determination of the optimal phase angle cut-off value at ICU admission using maximally se-lected rank statistics. X-axis: Phase Angle 1 (degrees), ranging roughly from 4.7 to 6.7; Y-axis: Density (i.e., how fre-quently different Phase Angle values occur in each group); (Blue curve: Density estimate for the ‘low’ group, Red curve: Density estimate for the ‘high’ group).

**Figure 2 healthcare-13-01463-f002:**
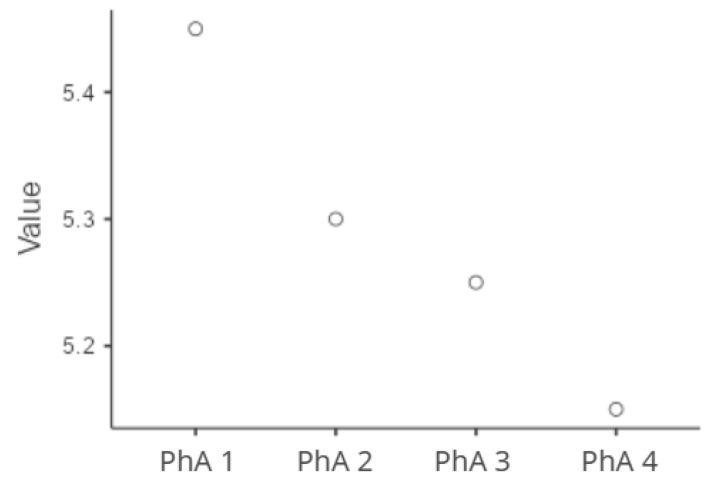
Longitudinal PhA changes.

**Figure 3 healthcare-13-01463-f003:**
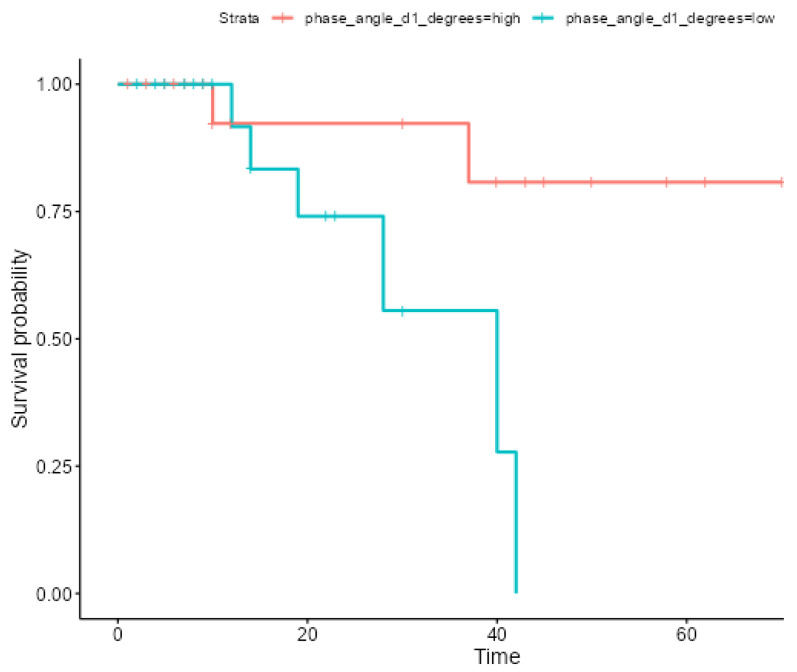
Kaplan–Meier survival curves for ICU patients based on the cut-off (PhA = 5.4°).

**Figure 4 healthcare-13-01463-f004:**
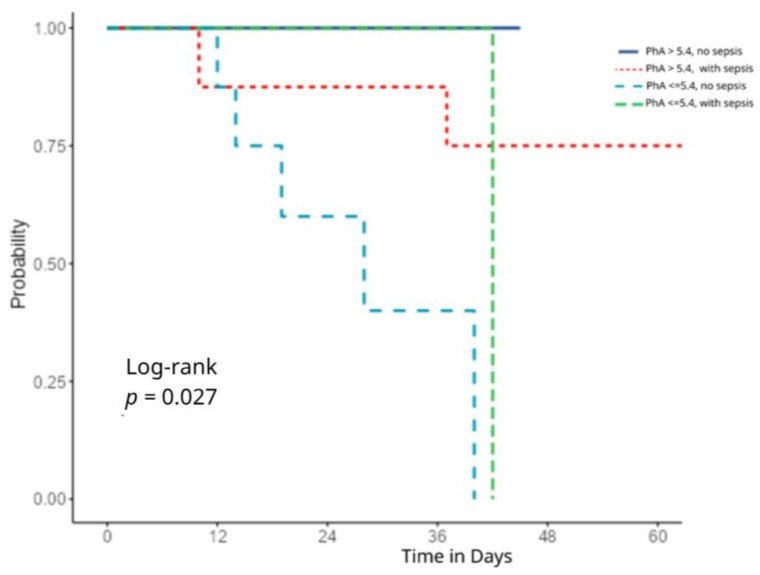
Kaplan–Meier survival curves stratified by phase angle and sepsis status.

**Table 1 healthcare-13-01463-t001:** PICO Criteria.

Parameter	Criteria
Participants	Adult critically ill patients admitted to the ICU at a tertiary hospital (Evangelismos Athens General Hospital), between May and November 2024.
Intervention/Exposure	Assessment of phase angle (PhA) on ICU admission and its changes over time
Comparison	Patients stratified by PhA value: PhA ≤ 5.4° (lower phase angle group) PhA > 5.4° (higher phase angle group)
Outcomes	Primary outcome: Association between PhA at ICU admission and in-hospital mortality. Secondary outcomes: Association of PhA with clinical severity scores (APACHE II and SOFA) on ICU admission; relationship between PhA and anthropometric/metabolic parameters; longitudinal changes in PhA during ICU stay, analyzed across four time-points; impact of PhA decline over time on survival outcomes; sepsis status

**Table 2 healthcare-13-01463-t002:** Phase angle correlations on ICU admission ^1^.

	Phase Angle ≤ 5.4 (*n* = 22)	Phase Angle > 5.4 (*n* = 21)	*p*
Age, years	68 (13)	49 (25)	<0.001 ^2^
Body Height, m	1.70 (0.16)	1.78 (0.07)	0.08 ^2^
Body Weight, kg	79.5 (16.5)	82 (22)	0.80 ^2^
BMI, kg/m^2^	26.5 (3.33)	25.7 (4.1)	0.11 ^2^
Body Fat, kg	39.36 (12.7)	31.45 (14.5)	0.08 ^3^
Fat Free Mass, kg	36.0 (2.75)	44.0 (5.0)	<0.001 ^2^
Calf Circumference, cm	32.0 (4.75)	35.0 (1.0)	<0.001 ^2^
Extra Cellular Water, L	26.0 (2.75)	32.0 (3.0)	<0.001 ^2^
Mid-Upper Arm Muscle Circumference, cm	38.0 (2.91)	42.0 (6.4)	0.009 ^2^
Resting Energy Expenditure, kcal	1856 (560)	2082 (377)	0.07 ^2^
REE, kcal/kg	23.13 (3.02)	27.40 (5.88)	0.002 ^2^
Respiratory Quotient, %	0.82 (0.06)	0.83 (0.04)	0.24 ^2^
APACHE II Score	10 (9.5)	11 (8.0)	0.97 ^2^
SOFA Score	6.0 (5.7)	7.5 (7.0)	0.41 ^2^
Medication Number	9.73 (2.88)	9.05 (3.46)	0.65 ^3^
Dyslipidemia	8 (38.1)	2 (9.5)	0.03 ^4^
Hypertension	8 (38.1)	5 (23.8)	0.317 ^4^
Diabetes	3 (14.3	1 (4.8)	0.29 ^4^
Coronary Disease	3 (14.3)	2 (9.5)	0.63 ^4^
COPD	2 (9.5)	1 (4.8)	0.54 ^4^
Chronic Renal Failure	0	0	-
Hepatic Failure	0	1 (4.8)	0.31 ^4^
Cancer	2 (9.5)	0	0.14 ^4^
COVID-19	1 (4.5)	1 (4.8)	0.97 ^4^

^1^ Data are displayed as mean (SD, column %) and median (interquartile range). ^2^ Mann–Whitney U test. ^3^ Independent samples *t*-test. ^4^ χ^2^ test.

**Table 3 healthcare-13-01463-t003:** Hazard Ratios from multivariable cox regression for predictors of ICU outcome (mortality) in 43 critically ill patients.

Independent Variables	HR	95% CI	*p*
Sex	0.16	0–18.4	0.45
APACHE	1.59	1.01–2.50	0.04
SOFA	0.54	0.24–1.22	0.13
PhA	0.10	0.01–0.76	0.02
Sepsis	2.92	0.36–24.05	0.31
Body fat (kg)	0.84	0.68–1.03	0.09
Medication	0.99	0.53–1.87	0.98

## Data Availability

Data are available from the corresponding author upon reasonable request.

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
