# Peer review of "Phase Angle Trajectory Among Critical Care Patients: Longitudinal Decline Predicts Mortality Independent of Clinical Severity Scores"

_healthcare, 2025, doi:10.3390/healthcare13121463_

Round 1

Reviewer 1 Report

Comments and Suggestions for Authors

Dear Authors 

Thank you for sending this manuscript for publication. The study supports the predictive ability of PhA towards ICU mortality. While the paper is written in good language, I have raised some recommendations to increase the likelihood of publication as follows:

1. Establishing a cutoff point of PhA (below and above 5.4) is still questionable and needs further clarification and support from the literature. What is the problem if you left PhA as a continuous variable rather than making it a dichotomous-like measurement? 

2. I’m still uncertain what patients’ etiologies were on admission. More description about these 43 patients is required.

3. Why specifically were sepsis patients used to estimate their relationships with PhA, and survival? What about other conditions?

4. PhA and its association with both SOFA and APACHE II scores are acknowledged. However, because BIA technology is based on body composition, what is your impression about the impact of nutritional failure or malnutrition risks, which may be associated with lower survival?

5. I may suggest creating simpler figures with better resolution if possible.

6. The discussion part needs to be expanded with much support from existing literature.

Good luck  

Author Response

Reviewer # 1

Comment 1: Establishing a cutoff point of PhA (below and above 5.4) is still questionable and needs further clarification and support from the literature. What is the problem if you left PhA as a continuous variable rather than making it a dichotomous-like measurement?

Response 1: Thank you for pointing this out. We agree with this comment. Firstly, we tracked longitudinal changes, using Durbin-Conover pairwise comparisons and we observed a significant decline in PhA over time, (considering PhA as a continuous variable, see Fig.2 in the revised MS ). In addition, following your comment, we conducted a multivariable survival analysis, and PhA was entered into the model as a continuous variable, and it was found to be significantly associated with survival (Page 8, line 213 in the revised MS)

Comment 2: I’m still uncertain what patients’ etiologies were on admission. More description about these 43 patients is required.

Response 2: Thank you for this comment. We have revised the results section and emphasized this point on lines 171-174 in the revised MS.

Comment 3: Why specifically were sepsis patients used to estimate their relationships with PhA, and survival? What about other conditions?

Response 3: Thank you for this comment. Patients with all included conditions -and not only limited to sepsis- were used to estimate the relationships with PhA, and survival.

Comment 4: PhA and its association with both SOFA and APACHE II scores are acknowledged. However, because BIA technology is based on body composition, what is your impression about the impact of nutritional failure or malnutrition risks, which may be associated with lower survival?

Response 4: Thank you for this comment. We agree that bioelectrical impedance analysis, and particularly PhA, reflects not only illness severity but also underlying nutritional and cellular integrity. In our cohort, patients with lower PhA values also had significantly lower fat-free mass, muscle circumference, and other anthropometric indicators, suggesting a higher risk of malnutrition or muscle wasting. According to the European Society for Clinical Guidelines for ICU patients, every critically ill patient staying for more than 48 hours in the ICU is considered to be at risk for malnutrition. Thus, malnutrition risk was uniformly present and therefore not a confounding factor in the association between PhA and survival. We have now expanded the discussion to address this link and emphasize the need for future studies that incorporate both BIA-derived parameters and validated nutritional risk assessments to clarify their combined prognostic value (page 2, lines 51-53 in the revised MS).

Comments 5: I may suggest creating simpler figures with better resolution if possible.

Response 5: Thank you for this comment. We have updated our figures, and changed Fig. 3 (page 7, line 200 in the revised MS).

Comments 6: The discussion part needs to be expanded with much support from existing literature.

Response 6: Thank you for this comment. We have expanded the discussion part.

Reviewer 2 Report

Comments and Suggestions for Authors

Abstract

Small sample size (n=43) not mentioned or addressed.

Lack of detail on PhA measurement frequency.

No emphasis on the clinical utility or integration potential of PhA beyond existing scores.

Introduction

Repetitive explanation of phase angle and its physiological basis.

Limited critical evaluation of the shortcomings of SOFA and APACHE II.

Does not clearly define the knowledge gap this study fills.

Materials and Methods

Timing of PhA measurements is inconsistent and vaguely described.

No power calculation or justification for sample size.

Data availability statement is incomplete with placeholder text.

Exclusion criteria such as “extensive edema” may introduce subjectivity without further definition.

Results

Dense and hard-to-read table formatting.

Minimal exploration of underlying mechanisms or causality in PhA-mortality relationship.

Kaplan-Meier and Cox results are presented without nuanced interpretation over time.

Some key variables (e.g., comorbidities, medications) not reported or adjusted for.

Discussion

Redundant content from the introduction.

Limited critique of the performance and limitations of SOFA/APACHE II in context.

Underexplored practical challenges of PhA application in ICU settings.

Does not discuss confounders or other biomarkers that may influence PhA.

Limitations

No mention of measurement variability or operator dependence in BIA.

Omits discussion of potential confounders (e.g., fluid therapy, inflammation, medications).

Single-center design and lack of blinding not highlighted.

Conclusion

Overly cautious tone fails to emphasize the novel aspect of longitudinal PhA tracking.

Lacks concrete suggestions for how to implement findings into clinical practice.

Author Response

Reviewer # 2

Comment 1: Abstract: Small sample size (n=43) not mentioned or addressed.

Lack of detail on PhA measurement frequency.

No emphasis on the clinical utility or integration potential of PhA beyond existing scores.

Response 1: Thank you for pointing this out. We agree with this comment. Therefore, we have made the necessary additions to the abstract (page 1, lines 23-24, 36 in the revised MS).

Comment 2: Introduction: Repetitive explanation of phase angle and its physiological basis.

Limited critical evaluation of the shortcomings of SOFA and APACHE II.

Does not clearly define the knowledge gap this study fills.

Response 2: Thank you for this comment. We have, accordingly, made amendments on the introduction part (Page 2, lines 67-77 in the revised MS).

Comment 3: Materials and Methods: Timing of PhA measurements is inconsistent and vaguely described.

No power calculation or justification for sample size.

Data availability statement is incomplete with placeholder text.

Exclusion criteria such as “extensive edema” may introduce subjectivity without further definition.

Response 3: The part concerning the timing of PhA measurements has been revised (page 3, line 107 in the revised MS). Power calculation for sample size has been included (page 5, line 159-167 in the revised MS). “Extensive edema” has been re-expressed as “edema” (page 3, line 91 in the revised MS).

Comment 4: Results: Dense and hard-to-read table formatting.

Minimal exploration of underlying mechanisms or causality in PhA-mortality relationship.

Kaplan-Meier and Cox results are presented without nuanced interpretation over time.

Some key variables (e.g., comorbidities, medications) not reported or adjusted for.

Response 4: Thank you for this comment. Given the great number of variables, we tried to summarize all the data in one table, showcasing all the tests performed, in order to avoid overwhelming the reader.

Underlying mechanisms are included in the discussion section (page 9, lines 284-298 in the revised MS)

Kaplan-Meier and Cox results have been updated (page 6, lines 194-198 and page 7, 201-210 in the revised MS)

Comorbidities and medications are now reported on Table 2 (page 6, line 175 in the revised MS). Medications have been adjusted for. However, given that only a small number of participants was included in each comorbidity category, the results would not be representative in our cohort.

Comments 5:

Limitations: No mention of measurement variability or operator dependence in BIA.

Omits discussion of potential confounders (e.g., fluid therapy, inflammation, medications).

Single-center design and lack of blinding not highlighted.

Response 5: Thank you for this comment. Measurement has been conducted by one trained dietitian, as has now been specified (page 3, line 107 in the revised MS). Blinding was not feasible, provided the study design. Discussion of potential confounders is added (page 10, lines 303-309 in the revised MS).

Comment 6:

Conclusion: Overly cautious tone fails to emphasize the novel aspect of longitudinal PhA tracking.

Lacks concrete suggestions for how to implement findings into clinical practice.

Response 6: Thank you for this comment. A conclusions section has been added.

Reviewer 3 Report

Comments and Suggestions for Authors

METHODS

The sample size (n=43) is relatively small, which may limit generalizability. While this is acknowledged, it should be discussed further with respect to power calculations or effect size estimates.

Although adjustments for APACHE II and sepsis were made, other potential confounders (e.g., renal failure, mechanical ventilation, corticosteroid use) that affect body composition or fluid status are not discussed.

The cut-off of 5.4° was chosen using maximally selected rank statistics, but external validation or comparison to previously established cut-offs (e.g., 4.8°) should be discussed in more detail.

Within the methods section I would either include a link to another paper where the methods are explicitly described or include a supplementary appendix on the details of calculating BIA and PhA. Given this is a more novel modality being measured many readers will be unfamiliar with it. 

Author Response

Reviewer # 3

Comment 1: The sample size (n=43) is relatively small, which may limit generalizability. While this is acknowledged, it should be discussed further with respect to power calculations or effect size estimates.

Response 1: Thank you for this comment. Power calculation for sample size has been included (page 5, line 160-167 in the revised MS).

Comment 2: Although adjustments for APACHE II and sepsis were made, other potential confounders (e.g., renal failure, mechanical ventilation, corticosteroid use) that affect body composition or fluid status are not discussed.

Response 2: Corticosteroid use was an exclusion criterion (page 3, line 94 in the revised MS). Other potential confounders, such as medication, and body fat has been adjusted for.

Comment 3: The cut-off of 5.4° was chosen using maximally selected rank statistics, but external validation or comparison to previously established cut-offs (e.g., 4.8°) should be discussed in more detail.

Response 3: Thank you for this comment. PhA has also been considered as a continuous variable in the multivariable analysis, to avoid subjectivity (page 4, line 117 in the revised MS).

Comment 4: Within the methods section I would either include a link to another paper where the methods are explicitly described or include a supplementary appendix on the details of calculating BIA and PhA. Given this is a more novel modality being measured many readers will be unfamiliar with it. 

Response 4: Thank you for this comment. We have included a reference leading to a paper explaining the procedure of BIA (page 4, line 137 in the revised MS)

Round 2

Reviewer 1 Report

Comments and Suggestions for Authors

Thank you for submitting this revision. I feel that you sufficiently addressed the points raised in the previous revision. Thank you

Reviewer 2 Report

Comments and Suggestions for Authors

I have no more comments